# Quantifying the indirect impact of COVID-19 pandemic on utilisation of outpatient and immunisation services in Kenya: a longitudinal study using interrupted time series analysis

Steven Wambua [1], Lucas Malla,[2] George Mbevi,[2] Joel Kandiah,[3] Amen-Patrick Nwosu [4], Timothy Tuti [2], Chris Paton,[4] Bernard Wambu,[5] Mike English [2,4], Emelda A Okiro[1,4]

For numbered affiliations see end of article.

**Correspondence to**
Mr Steven Wambua;
stevenwambua3@gmail.com

## ABSTRACT

**Objective**  In this study, we assess the indirect impact of COVID-19 on utilisation of immunisation and outpatient services in Kenya.

**Design**  Longitudinal study.

**Setting**  Data were analysed from all healthcare facilities reporting to Kenya's health information system from January 2018 to March 2021. Multiple imputation was used to address missing data, interrupted time series analysis was used to quantify the changes in utilisation of services and sensitivity analysis was carried out to assess robustness of estimates.

**Exposure of interest**  COVID-19 outbreak and associated interventions.

**Outcome measures**  Monthly attendance to health facilities. We assessed changes in immunisation and various outpatient services nationally.

**Results**  Before the first case of COVID-19 and pursuant intervention measures in March 2020, uptake of health services was consistent with historical levels. There was significant drops in attendance (level changes) in April 2020 for overall outpatient visits for under-fives (rate ratio, RR 0.50, 95% CI 0.44 to 0.57), under-fives with pneumonia (RR 0.43, 95% CI 0.38 to 0.47), overall over-five visits (RR 0.65, 95% CI 0.57 to 0.75), over-fives with pneumonia (RR 0.62, 95% CI 0.55 to 0.70), fourth antenatal care visit (RR 0.86, 95% CI 0.80 to 0.93), total hypertension (RR 0.89, 95% CI 0.82 to 0.96), diabetes cases (RR 0.95 95% CI, 0.93 to 0.97) and HIV testing (RR 0.97, 95% CI 0.94 to 0.99). Immunisation services, first antenatal care visits, new cases of hypertension and diabetes were not affected. The post-COVID-19 trend was increasing, with more recent data suggesting reversal of effects and health services reverting to expected levels as of March 2021.

**Conclusion**  COVID-19 pandemic has had varied indirect effects on utilisation of health services in Kenya. There is need for proactive and targeted interventions to reverse these effects as part of the pandemic's response to avert non-COVID-19 indirect mortality.

### Strengths and limitations of this study

► This analysis is strengthened by use of a broad set of health services indicators and over a large number of health facilities nationally and a longer time period (39 months) allowing for the adjustment of pre-COVID-19 trends.

► We have adjusted for factors such as health workers strikes and missing data in the analysis strengthening the validity of the results.

► Data were analysed across the whole healthcare system in Kenya (both public and private sector), therefore, can be used to predict impact in other similar settings.

► COVID-19 outbreak and associated public health measures were not random. Other concurrent unmeasured factors or shocks could have contributed, however, small to the changes.

► This study does not allow for in-depth evaluation of the specific causes of the trends observed within a qualitative framework because it was purely quantitative.

## INTRODUCTION

The novel COVID-19 outbreak was declared a global pandemic by WHO on 11 March 2020. By 6 May 2021, 156 million cases and 3.2 million deaths have been reported globally.[1] Since the first case of COVID-19 was reported in Kenya on 13 March 2020, 1 62 098 cases and 2850 deaths were reported by 6 May 2021.[1] The government, in attempt to control the spread of the pandemic, instituted a raft of interventions. Consequently, beyond the pandemic's direct impact on the population health, indirect effects due to the control measures, changes in public and clinician behaviour and health system reorganisation are likely to manifest in changes to utilisation of essential health services.

The country has experienced three waves of the pandemic.[2] The first wave peaked in July/August 2020,[3] and in March 2021, the country experienced the third wave with the highest daily cases recorded since the start of pandemic. Throughout this period, a series of public health measures have been instituted by government authorities such as restrictions in movement, international travel and suspension of gatherings in various public places. In 5 March 2021, the COVID-19 vaccination campaign targeting 1.02 million health workers and those above the age of 58 years was launched.[4]

The public health interventions are expected to have economic and social impacts such as reductions in manufacturing, access to employment and basic necessities.[5 6] Consequently, access and utilisation of essential health services are likely to be affected.[7] Early modelled predictions showed reductions in utilisation of health services.[8 9] In addition, studies during previous epidemics in sub-Saharan Africa reported a reduction in utilisation of essential health services during and after outbreaks.[10–14] Various population groups are likely to be affected differently, with children and women at a higher risk.[10 15] These interruptions in health service utilisation are raising concerns of increased morbidity and mortality for non-COVID-19 illnesses and especially for childcare services.[9] Although recent studies have reported variable impact of the pandemic on various health services, the impact on administration of vaccines and monitoring a broad set of essential services over a longer -observation period after the pandemic was announced by WHO has not been evaluated rigorously in Kenya.[16–19]

Using the Kenya's routine health information system implemented through the District Health Information Software version 2 (DHIS 2), a database where all health facilities in Kenya are expected to report services they offered in a given month, this study aimed to assess the indirect impact of COVID-19 on utilisation of varied basic essential health services nationally.

## METHODS
### Timeline of events
#### Pre-COVID-19 measures
Two months before the first case of COVID-19 was reported in Kenya, the government increased preparedness towards the pandemic. The preparedness measures included monitoring suspected cases of COVID-19 at points of entry to the country, increasing capacity for testing and isolation centres, providing healthcare workers with information and tools for dealing with COVID-19 cases and enactment of an emergency response committee.[20 21]

#### Post-COVID-19 control measures
*Control measures to manage COVID-19 spread were first enacted on 13 March 2020*[22]
These were suspension of public gatherings including places of worship and limiting the number of people attending weddings and funerals. Institution of learning, bars and restaurants were also closed. Travel restrictions into and out of the country were put in place and the national dusk-to-dawn curfew was introduced. A month later, restrictions in movement into and out of counties with highest cases of COVID-19 were instituted and restaurants resumed operations under strict guidelines of social and physical distancing, temperature checks when accessing the restaurants and handwashing. In the month of May 2020, the government ceased movement into and out of the country through two neighbouring countries (Tanzania and Somalia). Home-based care was introduced for patients with COVID-19 in June 2020 and in July 2020 the government started relaxing restrictions on movement and local air travel and phased reopening of churches and other places of worship. In August 2020, international air travel resumed and in September 2020, operation of bars resumed. This was followed by phased reopening of schools and lifting of suspension on political gatherings in October 2020 and November 2020, respectively. Between December 2020 to February 2021, there was a national health workers strike triggered by demands for better working conditions such as provision of adequate personal protective equipment, enhanced risk allowances and a health insurance cover. Although the length of the strike varied by health facilities and cadre of health workers, we could not obtain a database which tracks strikes nationally, and we therefore assumed most of the health facilities were on strike during the whole period. All schools reopened in January 2021. The timeline of COVID-19 control measures is presented in figure 1.

### Data
#### Data sources
##### District Health Information Software version 2
DHIS2 is an open-source software platform for data reporting by all the health facilities in a country. The primary goals of the system were to establish a centralised database with reporting capabilities at health facilities, to define and determine the standards for local and national health service reports and to connect service delivery and other health system input databases.[23] Monthly aggregated hospital level data can be entered into the system using a variety of tools, including desktop computers, laptops, tablets and smartphones by health records and information officers (HRIOs) situated in various hospitals. For health facilities without a HRIO, data is sent to a central administrative unit where the data is aggregated and entered into the system. Strong technical capabilities, flexibility, cost-effectiveness, increased satisfaction and networking among stakeholders have been some of the strengths of DHIS2 reported in 11 countries.[24]

### Extracted data
We extracted monthly data from DHIS 2 for the period January 2018 to March 2021 on total outpatient visits (under and over-fives), the number of hypertension

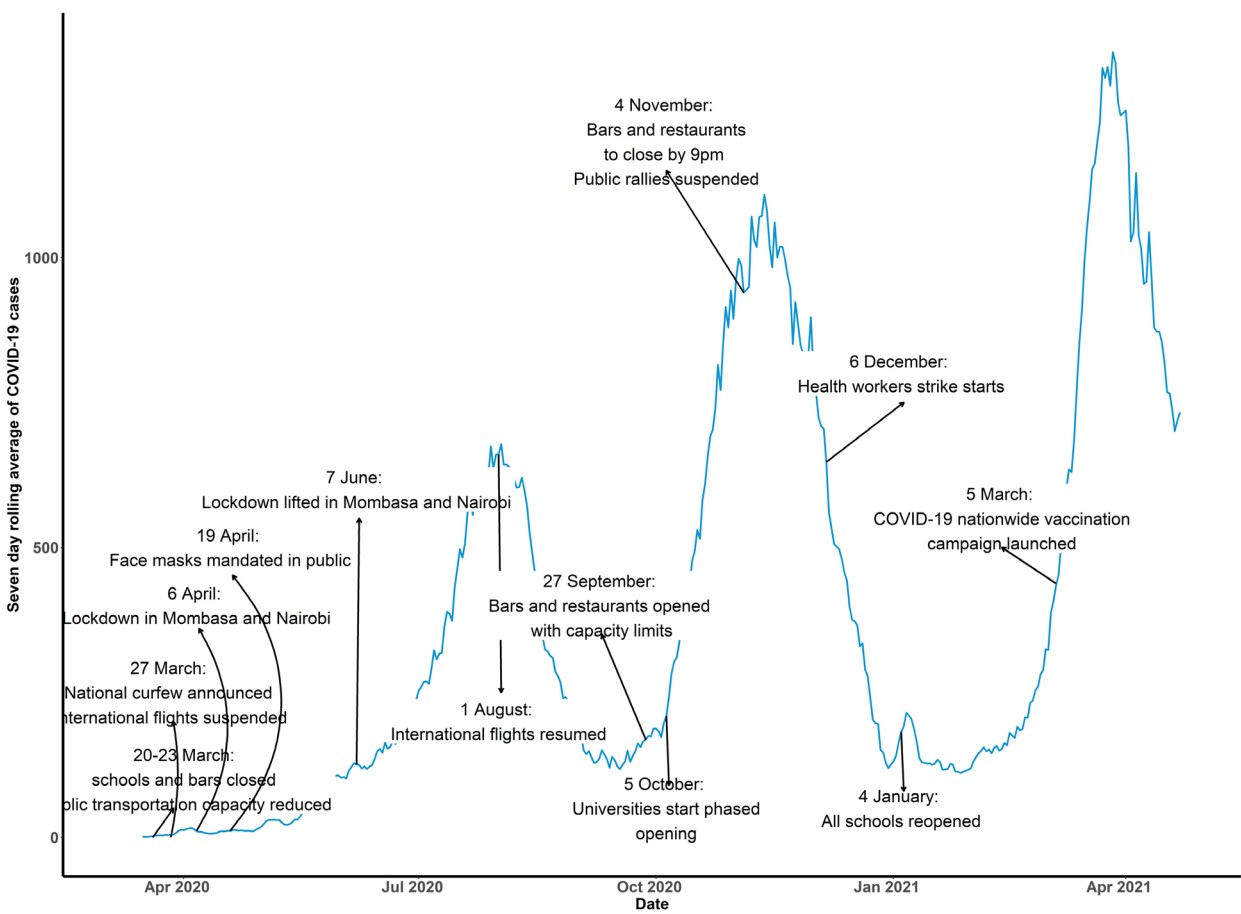

**Figure 1** Daily seven moving average trend of COVID-19 cases in Kenya showing various public health interventions initiated by the government to control the spread of the pandemic.

and diabetes cases and HIV tests performed, doses of immunisation antigens administered and antenatal care visits (the first (ANC 1) and fourth (ANC 4) visits). ANC 1 and ANC 4 are recommended by WHO as tracker indicators for ANC coverage and hence are reported in DHIS 2. A description of the indicators is presented in table 1.

Data were not available for the period January 2018 to September 2018 for hypertension and diabetes new cases. For both indicators and for relevant periods data were excluded from the analysis. We chose 2018 as a starting point because of prolonged healthcare worker strikes in 2017 which affected health services provision,[25] and consequently reporting. Data were cleaned to remove duplicated health facilities and those indicated as closed. Extreme outliers, defined as values that are more than three SD from the mean of reported values for a given health facility,[26 27] were identified, investigated and treated as missing. For each health facility, we obtained the administrative units, level of the facility (level 2: dispensaries with outpatient services only, level 3: comprehensive primary healthcare facilities, level 4: primary referral hospitals, level 5: secondary referral hospitals and level 6: national teaching and referral hospitals) and whether the health facility is private or public.

## Statistical analysis
### Missing data in DHIS 2
Missing data occurred for the indicators in a given month for a given health facility. Missingness varies by health facility and consistency in reporting overtime. Incompleteness in reports has been attributed to inadequate human resources, frequent power outages and slow internet connectivity, use of manual and electronic systems concurrently and frequent changes in DHIS 2 versions.[28] Strategies to improve reporting such as improving clinical care documentation, motivation among staff, government commitment and extensive donor support have been identified as strategies to improve completeness in DHIS 2.[29 30]

### Handling missing data
To adjust for incompleteness in reporting, multiple imputation (MI) was performed.[31–33] MI has been shown to perform better in handling missing data in comparison to other methods.[34] Missing monthly values were imputed using a mixed effects model in a joint modelling framework.[35 36] Health facility ownership (public or private), level of health facility, time (month and year) and COVID-19 binary indicator (0—months before pandemic and 1—months post pandemic) were used as covariates

**Table 1** Description of indicators analysed in this study and Kenyan Ministry of Health (MOH) source forms used to capture the data

| Category | Description | Assigned names in this study | Source form |
|---|---|---|---|
| Immunisation | Bacillus Calmette-Guerin (BCG) vaccine doses administered | BCG | MOH 710 |
| | Oral polio vaccine (OPV) doses administered | OPV dose 1, dose 2 and dose 3 | MOH 710 |
| | Rotavirus vaccine doses administered | Rotavirus dose 1 and dose 2 | MOH 710 |
| | Pneumococcal conjugate vaccine doses administered | Pneumococcal dose 1, dose 2 and dose 3 | MOH 710 |
| | Diphtheria, Tetanus, Pertussis (DPT) vaccine doses administered | DPT 1, 2 and 3 | MOH 710 |
| | Inactivated Polio Vaccine (IPV) doses administered | IPV | MOH 710 |
| | Measles vaccine doses administered | Measles dose 1 and dose 2 | MOH 710 |
| Outpatient visits | Antenatal care (ANC) first visit | ANC 1 | MOH 711 |
| | Antenatal care fourth visits | ANC 4 | MOH 711 |
| | Outpatient department (OPD) visits in under-fives | OPD <5 years | MOH 705A |
| | Outpatient department visits in over-fives | OPD >5 years | MOH 705B |
| | Outpatient department visits with pneumonia in under-fives | OPD pneumonia<5 years | MOH 705 A |
| | Outpatient department visits with pneumonia in over-fives | OPD pneumonia>5 years | MOH 705B |
| | No of new cases of diabetes | Diabetes new cases | MOH 705 A and B |
| | No of new plus revisits of diabetes cases | Diabetes total cases | MOH 705 A and B |
| | No of new hypertension cases | Hypertension new cases | MOH 705 A and B |
| | No of new plus revisits of hypertension cases | Hypertension total cases | MOH 705 A and B |
| | No of HIV tests performed | HIV tests performed | MOH 731 |

with the health facility as a clustering variable. MI was performed for health facilities with more than 30% of months reported (at least 12 months reported) to reduce uncertainty in imputed values and ensure generalisability of the estimates. The missing patterns for each indicator are presented in online supplemental file 1 SI figure 1. The MI model specification has been provided in online supplemental file 2. Additionally, through a simulation study we found MI performance and efficiency was best when imputing for health facilities with more than 30% of months reported. The number of health facilities analysed is presented in online supplemental file 3 SI table 1.

### Interrupted time series analysis
#### Exploratory analyses
Data were aggregated monthly for all health facilities. Trends were plotted to visualise changes in utilisation of health services. Statistical process control (SPC) charts with the 2018–2019 average as a baseline were used to identify significant shifts in monthly values for 2020–2021. Values that are more than three SD from the mean are considered significant shifts and were carried forward for interrupted time series analysis.[37] Multiple change point analysis was applied to assess the influence of health worker strike on provision of health services.[38 39]

#### Segmented regression
We conducted interrupted time series analyses using monthly attendance counts for each indicator as outcomes. The period running from January 2018 to March 2020 when the first case was identified was defined as pre-COVID-19 and April 2020 to March

2021 as post-COVID-19. For indicators where changes were observed in SPC analysis, segmented regression were performed to model attendance before and after COVID-19 was reported.[40 41] The following equation specifies the model[40]:

$$\log\left(Y_t\right) = \beta_0 + \beta_1 * time_t + \beta_2 * COVID19_t + \beta_3 * time\ after\ COVID19_t$$

Where, $Y_t$ is the attendance in month $t$; *time* is a continuous indicator of time in months from January 2018; *COVID*19 is an indicator of time $t$ occurring before ($COVID19 = 0$ or after ($COVID19 = 1$ the outbreak, which was implemented at April 2021 in the series; and *time after COVID*19 is a continuous variable of the number of months after COVID-19 at time $t$. In the model, $\beta_0$ estimates the baseline level of attendance at time zero; $\beta_1$ estimates the change in monthly number of visits before COVID-19 (pre-existing trend); $\beta_2$ estimates the level change immediately after COVID-19 outbreak; $\beta_3$ estimates the change in the trend after COVID-19, compared with the pre-existing trend. A change in intercept (immediate COVID-19 effect) and change in slope (gradual COVID-19 effect) were hypothesised.[41]

A generalised linear model was applied assuming a negative binomial distribution. The negative binomial model was selected due to variations in attendance at health facility level. The intraclass correlation coefficients for each indicator are provided in online supplemental file 4 table 1. We fitted two negative binomial models to account for over-dispersion, one without accounting for seasonality and another accounting for seasonality.[41–43] Model performance was evaluated using the Akaike's

information criterion.[44] Model checking was conducted for autocorrelation using the Durbin-Watson statistic and autoregressive moving average (ARMA) models were fitted for indicators with serial autocorrelation.[45–47] The ARMA model fitted is presented below;

$$X_t = c + \varepsilon_t + \sum_{j=1}^{p} \varphi_i \, X_{t-i} + \sum_{j=1}^{q} \theta_i \, \varepsilon_{t-i}$$

Where $\varphi$ is the AR model parameters, $\theta$ is the MA model parameters, c a constant and $\varepsilon$ is the error term. We fitted the ARMA model using various combinations of $p$ and $q$ and selected the model with the lowest Akaike information criterion (AIC). The *gcmr* package was used to implement the ARMA models.[48] Seasonality was adjusted using Fourier terms by specifying the sine and cosine pairs as 2 and the length of the period as 12 as recommended [41 49] Results were pooled across the multiple imputed datasets using Rubin's rules.[50] The negative binomial model, which was adjusted for seasonality was the best fitting model and its results are presented in this study. AIC values and the estimates from the negative binomial model where seasonality was not accounted for are provided in online supplemental file 4 table 2 and 3.

As a form of sensitivity analysis, we fitted models excluding months when the national strike occurred and compared estimates with those where data included the strike. We also fitted health-facility level generalised estimating equations (GEE) to test the impact of varying model assumptions on the primary model estimates and hence evaluate robustness of our results.[41]

Statistical significance was defined as p<0.05. All analyses were performed using R (V.3.6.3).

## Patient and public involvement

No patients were involved in this study. We have used secondary aggregated routine health information data available online.

## RESULTS
## COVID-19 impact

Annual trends show the first ANC visits remained unaffected while the fourth visits experienced a downward trend from March 2020. Immunisation services remained unaffected with observed spikes in administration of measles vaccines in March 2020. Utilisation of outpatient services (overall and due to pneumonia) by under fives experienced drops after March 2020. Reductions were also experienced in over-fives attendance, hypertension cases and diabetes attendance. HIV testing experienced a gradual decline over the years (figure 2).

Further, SPC charts confirmed significant reductions (less than 3 SD) in ANC 4 starting April 2020. Immunisation services remained unaffected during the same period, with significant increase (more than 3 SD) in measles vaccination in March 2020. Moreover, significant reductions in under-fives attendance, over-fives attendance and new visits by hypertensive patients were observed starting

April 2020 with no significant reductions for HIV testing and diabetes visits (figure 3). Additionally, utilisation of most services reduced the most in December 2020 coinciding with start of healthcare workers strike, after which utilisation of most services started to go back to expected levels.

We fitted interrupted time series models for indicators that showed significant changes from the SPC charts. The rate ratios from the model are presented in table 1. The month-to-month changes before COVID-19 were generally increasing across all the indicators. There was an immediate statistically significant reduction in all the indicators post-COVID-19, in the month immediately after first case, except for ANC 1 and new cases of diabetes and hypertension, which were unaffected. The statistically significant level changes post-COVID-19 were outpatient attendance for children under-fives which reduced by 50%, those for outpatients' over-fives by 35%, under-fives pneumonia outpatients by 43%, over-fives pneumonia outpatients by 38%, ANC fourth visit by 14%, total cases of diabetes by 5%, new cases of hypertension by 11% and HIV tests by 3%. There was a slight but statistically significant month-to-month increase in services post-COVID-19 (April 2020 to March 2021) of 5% for under-fives outpatients attendance, 2% for over-fives outpatients, 4% for under-fives pneumonia outpatients, 3% for over-fives pneumonia patients and no significant month-to-month changes for ANC visits, diabetes and hypertension cases. The trends from the fitted interrupted time series model are visually represented in figure 4.

## Sensitivity analyses

Change point analysis showed the health workers' strike, which started in December 2020 had a significant impact on ANC fourth visits, and no effect on the other indicators (online supplemental file 5 SI figure 1). Further, excluding the strike period (December 2020 to February 2021) from the segmented regression models of all indicators evaluated resulted in estimates that are not different from primary model estimates (online supplemental file 5 SI table 1). Estimates from the GEE models were not different from the primary model indicating robustness of reported estimates (online supplemental file 5 SI table 2).

## DISCUSSION

Using DHIS2 health facility-level monthly reported outpatient data, we provide evidence of COVID-19 impact on utilisation of basic health services in Kenya. The announcement of the first case of COVID-19 in Kenya in March 2020 and the intervention measures that followed coincided with sharp declines in outpatient and ANC fourth visits nationally. By the end of this study, health services are still in the process of returning to pre-COVID-19 levels. However, immunisation services remained unaffected.

**Table 2** Segmented regression results showing rate ratios (RR) for COVID-19 intervention, time (pre-existing trend) and trend (post-COVID-19 trend)

| Covariate | OPD <5 years | | | OPD >5 years | | | OPD pneumonia <5 years | | | OPD pneumonia >5 years | | |
|---|---|---|---|---|---|---|---|---|---|---|---|---|
| | RR | 95% CI | P value | RR | 95% CI | P value | RR | 95% CI | P value | RR | 95% CI | P value |
| COVID-19 | 0.50 | (0.44 to 0.57) | <0.01 | 0.65 | (0.57 to 0.75) | <0.01 | 0.43 | (0.38 to 0.47) | <0.01 | 0.62 | (0.55 to 0.70) | <0.01 |
| Time | 1.00 | (0.99 to 1.01) | 0.15 | 1.00 | (1.00 to 1.01) | 0.02 | 1.01 | (1.00 to 1.01) | <0.01 | 1.00 | (0.99 to 1.01) | 0.05 |
| Trend | 1.05 | (1.03 to 1.06) | <0.01 | 1.02 | (1.00 to 1.04) | 0.03 | 1.07 | (1.05 to 1.08) | <0.01 | 1.03 | (1.02 to 1.05) | <0.01 |

| Covariate | ANC 1 | | | ANC 4 | | | Diabetes new cases | | | Diabetes total cases | | |
|---|---|---|---|---|---|---|---|---|---|---|---|---|
| | RR* | 95% CI* | P value* | RR | 95% CI | P-value | RR | 95% CI | P value | RR | 95% CI | P value |
| COVID-19 | 0.96 | (0.83 to 1.10) | 0.55 | 0.86 | (0.80 to 0.93) | <0.01 | 1.17 | (0.89 to 1.52) | 0.25 | 0.95 | (0.93 to 0.97) | <0.01 |
| Time | 1.00 | (0.99 to 1.00) | 0.61 | 1.00 | (0.99 to 1.00) | 0.13 | 0.99 | (0.98 to 1.00) | 0.13 | 1.00 | (1.00 to 1.01) | <0.01 |
| Trend | 1.01 | (0.99 to 1.03) | 0.12 | 1.00 | (0.99 to 1.01) | 0.90 | 0.99 | (0.97 to 1.01) | 0.57 | 1.00 | (1.00 to 1.00) | 0.05 |

| Covariate | Hypertension new cases | | | Hypertension total cases | | | HIV tests Performed | | |
|---|---|---|---|---|---|---|---|---|---|
| | RR | 95% CI | P value | RR | 95% CI | P value | RR* | 95% CI* | P value* |
| COVID-19 | 0.87 | (0.75 to 1.00) | 0.05 | 0.89 | (0.82 to 0.96) | <0.01 | 0.97 | (0.94 to 0.99) | 0.01 |
| Time | 1.00 | (0.99 to 1.01) | 0.81 | 1.01 | (1.00 to 1.01) | <0.01 | 0.97 | (0.97 to 0.97) | <0.01 |
| Trend | 1.00 | (0.99 to 1.01) | 0.59 | 1.00 | (0.99 to 1.01) | 0.90 | 1.01 | (1.00 to 1.01) | <0.01 |

The 95% CIs and p values are also show.

*Estimate after fitting ARMA model to indicator where autocorrelation was detected. The ARMA (p, q) parameters for ANC 1 are (1,0) and (2,0) for HIV tests performed.

ANC, antenatal care ; ARMA, autoregressive moving average; OPD, outpatient department.

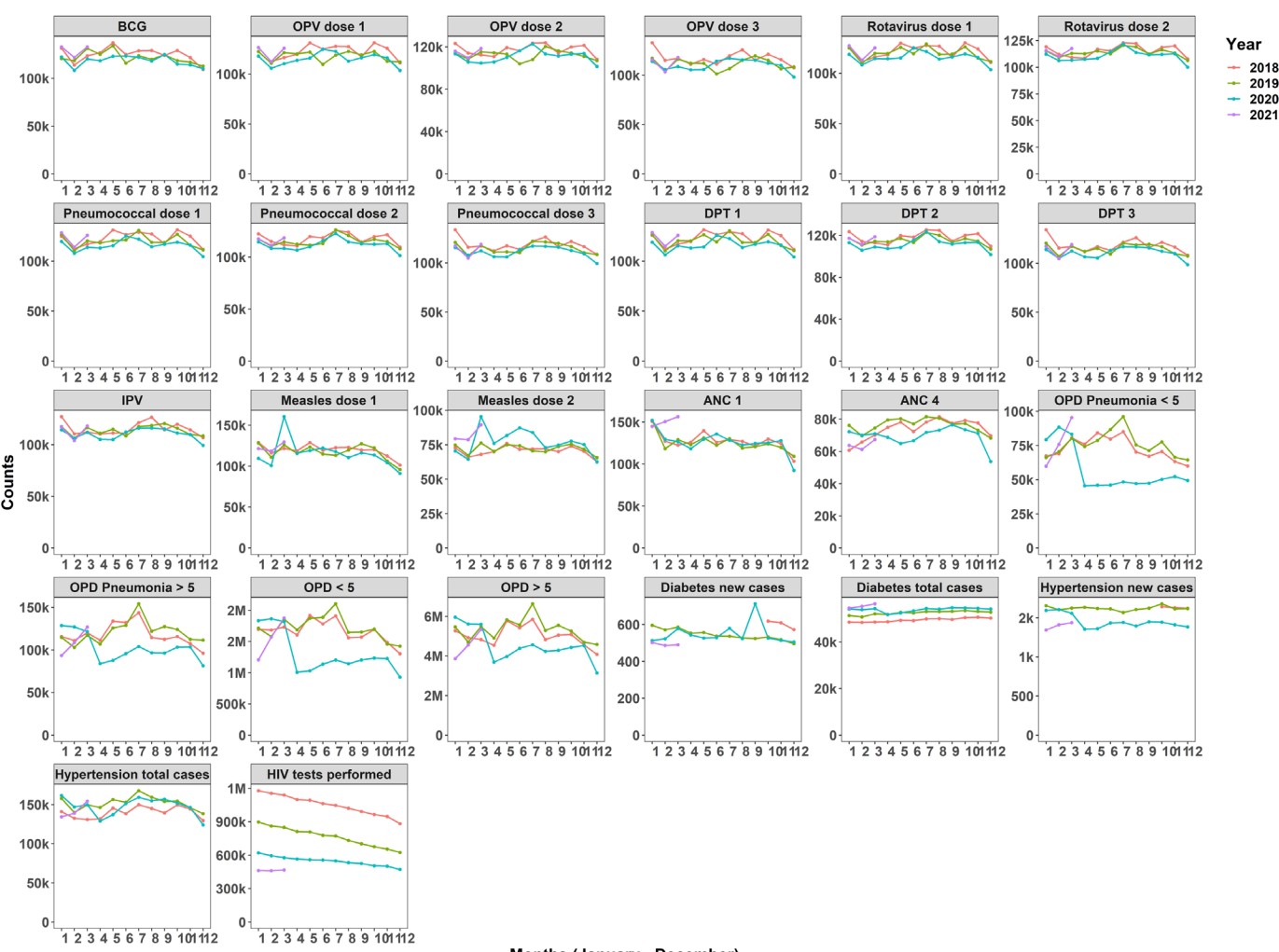

**Figure 2** Temporal trends in monthly immunisation and outpatient attendance nationally and by year. ANC, antenatal care; OPD, outpatient department;BCG,Bacillus Calmette–Guérin vaccine; OPV, Oral polio vaccine;DPT, Diphtheria, Tetanus, Pertussis vaccine; IPV, Inactivated Polio Vaccine

Previous studies have found variable impacts on immunisation services.[17 51 52] In two studies that evaluated performance of routine immunisation on selected indicators in Kenya, which used a relatively shorter period and didn't account for missing data, COVID-19 had no substantial impact on vaccination coverage, ANC first visits and a significant increase in measles immunisation in March 2020 was reported.[17 51] The significant increase in measles vaccines in March 2020 was due to increased immunisation to make up for stock-out of measles vaccines between November 2019 and January 2020.[17] The sustained immunisation levels in the other antigens suggests there were no significant disruption to vaccine supply chain resulting from the pandemic, and confirmed by the National Vaccines and Immunisation Programme (NVIP).[17] Additionally, where health facilities designated as vaccination centres were assigned as COVID-19 isolation centres, the vaccines programme moved immunisation services to neighbouring health facilities.[17] These strategies illuminate why immunisation services remained unaffected during the pandemic,

contrary to earlier predictions of reductions in immunisation.[8 9] Although not statistically significant, the slight reductions in the number of vaccines administered in December 2020 were likely attributed to the nationwide health worker strike, which led to staff shortages consequently affecting administration of the vaccines. These results strengthen previous findings with no observable differences in mean monthly number of immunisation and total ANC visits over a much shorter study period March–June 2020 relative to the same period in 2019 in Kenya.[52] Additionally, in a recent survey across 18 African countries, which evaluated disruption to essential health services in Africa during COVID-19, found that vaccination was the least disrupted service across all countries.[30] In summary, immunisation services were unaffected likely because of a number of reasons; the concerted effort by the NVIP to sustain supply of vaccines and unavailability of alternative sources for vaccination outside of the health system.

There were significant drops in nearly all outpatient services evaluated in this study. Total outpatient and

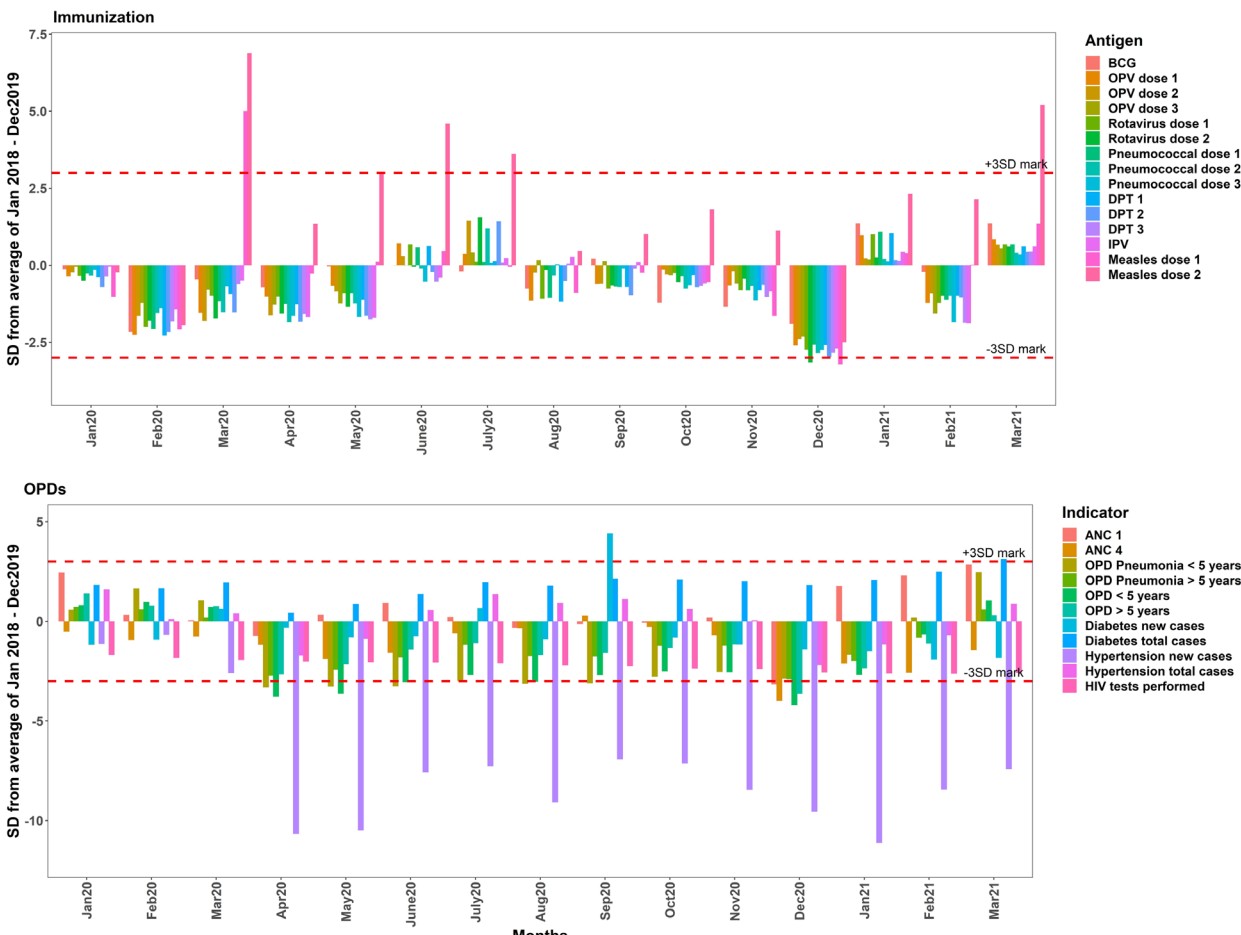

**Figure 3** Statistical process control chart of immunisation, antenatal care (ANC) and outpatient services. Horizontal dashed lines represent the 3 SD mark. OPD, outpatient department;BCG,Bacillus Calmette–Guérin vaccine; OPV, Oral polio vaccine;DPT, Diphtheria, Tetanus, Pertussis vaccine; IPV, Inactivated Polio Vaccine

pneumonia specific outpatient attendance were most affected, with utilisation of the services dropping by half for under-fives. Moreover, COVID-19 had an impact on ANC 4, total attendance for hypertension and diabetes and HIV testing. Similar findings have been reported in other low- and middle-income countries.[16 17 19 53–56] Studies evaluating the impact of lockdown measures to combat COVID-19 in South Africa observed a substantial drop in primary healthcare services utilisation.[16 55] Significant drops in essential health services were also experienced following institution of public health measures to combat COVID-19 in Kinshasa, Democratic Republic of Congo.[19] Disruptions in general attendance have also been reported in various studies globally.[53 57–60]

Various factors could explain the downward trends in specific outpatient services. In a survey conducted in Kenya to assess health services utilisation during COVID-19, common causes reported by respondents include fear of risk of catching coronavirus at health facilities (26%), reduced incomes affecting ability to meet transport costs and other healthcare related costs (17%), shortage of healthcare workers in health facilities (14%), difficulties in accessing health facilities due to lockdowns and curfew (14%) and closing of some health facilities (14%).[61] The

substantial declines for under-fives attendance are likely associated with reduced mixing due to closure of schools, improved hygiene practices and parents choosing to manage non-severe illnesses at home. Although attendance for ANC 4 was affected, it is unclear why the first visits were not affected. Notwithstanding, this might suggest that pregnant women attach greater importance to the primary ANC visit as has been reported,[62 63] and hence despite the prevailing conditions managed to prioritise at least one visit to a health facility. Additionally, data have suggested deliveries in health facilities were also not affected during the pandemic [64], and this likely suggests the population of pregnant women remained relatively comfortable to use health services despite the pandemic.

A survey in Ethiopia among diabetic and hypertensive patients reported unavailability, unaffordable or increased price of medications and interruptions in follow-up visits were common barriers to accessing chronic care units in public facilities during the pandemic.[65] Reduction in attendance for chronic conditions such as hypertensive cases is a significant finding as missing care for these chronic illnesses could lead to further complications and susceptibility to severe COVID-19,[66] and increased morbidity and mortality. The gradual decline

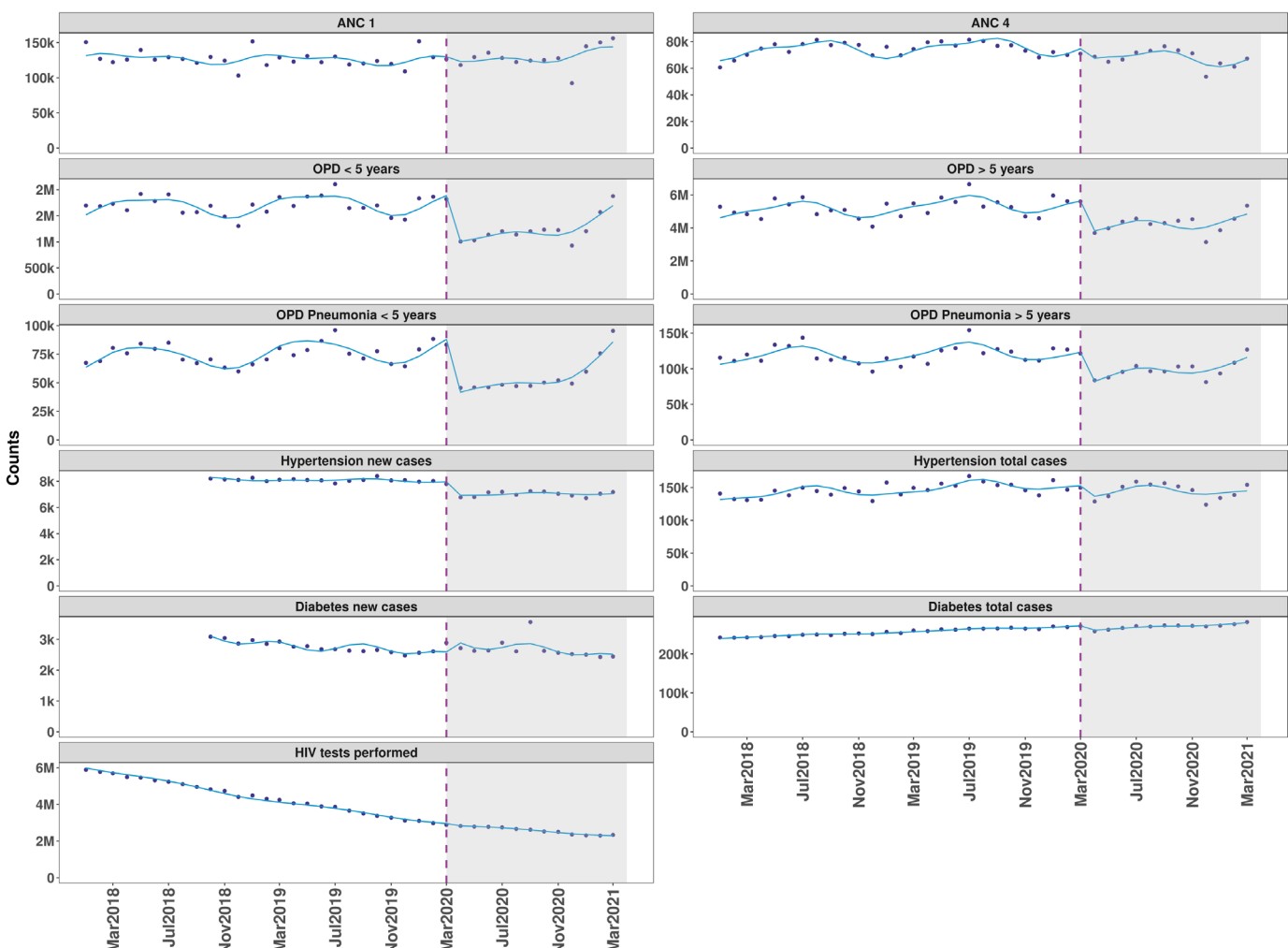

**Figure 4** Fitted lines of interrupted time series models for outpatient and antenatal care attendance. vertical lines represent the month (March 2020) COVID-19 was announced in Kenya and as a pandemic by the WHO. ANC, antenatal care; OPD, outpatient department.

in HIV testing pre-COVID-19 might suggest reduced coverage due to policies geared towards targeted testing as opposed to blanket testing.[67] Additionally increased uptake and accessibility to testing in pharmacies implemented in 2017 might be associated with reduced testing in health facilities.[68] Pre-existing challenges in access to health services such as poor road network, disruptions in supplies to health facilities, and limited or no capacity for domestic production of medical supplies could have compounded the dramatic downward trends in utilisation of outpatient services. Additionally, improved hand hygiene and use of face masks during the pandemic could have led to reduced risk of other infectious diseases and consequently fewer visits to health facilities.[69 70]

### Strengths and implications of the study

Although most of the public attention is on control measures of COVID-19, possible health consequences from the indirect effects of the measures should not be overlooked. We provide a comprehensive understanding of the present situation on utilisation of immunisation and outpatient services in Kenya. Although the findings

provide short-term estimates on the effect of COVID-19 at national level, studies could assess the long-term and differential effects at subnational level. We addressed possible confounders in assessing changes overtime. For instance, in line with a recent guide on using routine data to monitor the effects of COVID-19 by the WHO, we adjusted for missing data which would have affected the validity of the comparisons over time.[71] Additionally, incompleteness may lead to biased estimates and strategies to improve data quality in DHIS2 such as investment in better infrastructure, supervisory support, formal data quality assurance and human resources could improve reporting in Kenyan health facilities.[72 73] We also use sensitivity analysis to account for any uncertainty in the estimates due to other factors affecting utilisation of services such as healthcare workers strikes and health-facility specific variations, which reduced bias and improved precision of the estimates.

### Limitations and recommendations

In this study, controls were not used to differentiate the impact of COVID-19 from other possible causes of the

changes as most indicators were indirectly affected by the pandemic. However, since the drops in utilisation of services coincided with the introduction of COVID-19 intervention measures, the changes are attributed to COVID-19. We suggest sensitivity studies in future to assess any departures from the missing at random assumption when using MI for DHIS 2 data.

## CONCLUSION

In summary, COVID-19 pandemic has had varied indirect effects on utilisation of outpatient health services. Although utilisation of immunisation services remained unchanged, there was a significant negative impact on outpatient clinic and ANC visits nationally. Total outpatient attendances for children under-fives reduced by 50%, under-fives pneumonia presentations reduced by 50%, general over-five visits reduced by 35%, over-fives pneumonia reduced by 38%, ANC 4 visits reduced by 14%, total hypertension cases reduced by 11%, total diabetes cases reduced by 5% and HIV testing by 3%. There is need for proactive and targeted interventions to avert and reverse these effects in future pandemics. These include strict implementation of safe practices and infrastructural changes in health facilities to reassure the public that it is safe to go to health facilities. Other innovative measures such as safe modes of transport, mobile clinics and supplementary immunisation activities could be incorporated in the pandemic response to avert any negative effects on utilisation of essential health services.

**Author affiliations**
[1]Population Health Unit, KEMRI-Wellcome Trust Research Programme Nairobi, Nairobi, Kenya
[2]Health Services Unit, KEMRI-Wellcome Trust Research Programme Nairobi, Nairobi, Kenya
[3]Mathematics Institute, University of Warwick, Coventry, UK
[4]Nuffield Department of Clinical Medicine, Oxford Centre for Global Health Research, Oxford, UK
[5]Division of Neonatal and Child Health, Kenya Ministry of Health, Nairobi, Kenya

**Contributors** SW: conceptualisation; data curation; formal analysis; investigation; methodology; software; validation; visualisation; writing—original draft. LM: data curation; formal analysis; investigation; methodology; software; validation; visualisation; writing—review and editing. GM: data curation; investigation; software; validation; visualisation; writing—review and editing. A-PN: data curation; formal analysis; investigation; software; validation; visualisation; writing—review and editing. JK: data curation; formal analysis; investigation; software; validation; visualisation; writing—review and editing. TT: data curation; investigation; validation; writing—review and editing. CP: data curation; investigation; validation; writing—review and editing. ME: data curation; investigation; validation; writing—review and editing. EAO: conceptualization; data curation; investigation; validation; funding acquisition; writing—review and editing; guarantor.

**Funding** This research was funded in whole or in part by the Wellcome Trust Intermediate Fellow [Grant No. 201 866]. For the purpose of Open Access, the author has applied a CC-BY public copyright licence to any author accepted manuscript version arising from this submission.SW and EO are supported through a Wellcome Trust Intermediate Fellow (Grant No. 201 866). LM, JK, A-PN, GM, TT and CP are supported through Funds from the Wellcome Trust (Grant No. 207 522) awarded to Prof. ME as a senior Fellowship together with additional funds from a Wellcome Trust core grant awarded to the KEMRI-Wellcome Trust Research Programme (Grant No. 092654). SW, LM, GM, TT, ME and EAO acknowledge the support of the Wellcome Trust to the Kenya Major Overseas Programme (Grant No. 203 077).

**Competing interests** The authors declare that they have no competing interests

**Patient consent for publication** Not applicable.

**Ethics approval and consent to participate** The study does not contain any individual person's data.

**Provenance and peer review** Not commissioned; externally peer reviewed.

**Data availability statement** Data are available on reasonable request. Aggregated DHIS2 data are available online with access provided by Kenyan Ministry of Health https://hiskenya.org/dhis-web-commons/security/login.action.

**ORCID iDs**
Steven Wambua http://orcid.org/0000-0003-2300-7670
Amen-Patrick Nwosu http://orcid.org/0000-0001-6991-7798
Timothy Tuti http://orcid.org/0000-0002-7915-3004
Mike English http://orcid.org/0000-0002-7427-0826

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
