## [Reviewer comments · BMJ Open]

ARTICLE DETAILS

TITLE (PROVISIONAL)	Quantifying the indirect impact of COVID-19 pandemic on utilisation of outpatient and immunisation services in Kenya: A longitudinal study using interrupted time series analysis
AUTHORS	Wambua, Steven; Malla, Lucas; Mbevi, George; Kandiah, Joel; Nwosu, Amen-Patrick; Tuti, Timothy; Paton, Chris; Wambu, Bernard; English, Mike; Okiro, Emelda

VERSION 1 – REVIEW

REVIEWER	Di Bidino, Rossella Fondazione Policlinico Universitario Agostino Gemelli IRCCS, HTA Unit
REVIEW RETURNED	03-Aug-2021

GENERAL COMMENTS	The paper is clear on methods, in the description of the local reality, and confounding factors (ie strike, missing data). Sensitivity analysis confirmed the robustness of your work, I suggest minor revisions: - Some information is duplicated in Methods on pre e post COVID measures. Some of them are already reported in Introduction. Please, decide where to report it.- Figure 3 is not easy to read. Have you considered the option to put it in Appendixes?- In discussion, just introduce the point on potential impact of COVID 19 also on access to health services or mention it as point to investigate in future.
--

REVIEWER	Kumar, Arvind All India Institute of Medical Sciences, Department of Medicine
REVIEW RETURNED	08-Aug-2021

GENERAL COMMENTS	The authors have attempted to quantify the hindrance in the accessibility of the overall health facilities in Kenya health management systems due to the emergence of COVID-19. The study covers an important topic in the concerned area. However, the study requires revision before considering it for publication. 1) The abstract may be improved in terms of coverage of methodology part. The authors have used advanced statistical tools but they did not mention it in the abstract.2) Similarly, the methodology part in the main text should also have comprehensive section over statistical tools used in the study.
--

	3) In the methodology section, the independent variables and the dependent variables should be clearly defined for a thorough understanding of the readers. 4) Use of multiple imputation for handling missing data was not clearly mentioned in the manuscript. If there were no cases reported that does not mean the misreporting of the data. 5) Also, if multiple imputation is required, the authors should clearly mention its use in details. 6) The results of the study are not self-explanatory. For example, headings of the figures and tables are not clearly mentioned by the authors. 7) Overall, the structure of the manuscript should be revised and can be spell checked and corrected for grammatical errors to make it more comprehensive.
--	---

REVIEWER	Chandir, Subhash Harvard Medical School Center for Global Health Delivery–Dubai
REVIEW RETURNED	25-Aug-2021

GENERAL COMMENTS	The paper adds to the growing evidence on the impact of Covid-19 on essential health services. Their finds on the heterogeneous impact of Covid-19 on health services utility are exciting. Following are few suggestions for revision:  • Please consider adding references in "Pre-COVID-19 measures" and "Post-COVID-19 control measures" sections to support the narrative. • The study is based on dhis2 data. It would be helpful to provide information on the use of dhis2 system (reporting structure, frequency, data quality and reliability, monitoring, etc) to understand the comprehensiveness of data and its generalizability. There are many imputations, and although they are nicely carried out, please consider providing more context on the reason behind missing values. • A significant finding is no impact on ANC1 visit while significant reduction is reported for ANC4 visit. However, ANC2 and 3 visits were excluded from the analysis. Please consider providing context, like was it due to lack of data or some other factor? • The authors note there was a reduction in health visits for various reasons (pages 19 & 20). However, no explanation is given as to why the same deterrents did not affect the immunization visits. Moreover, it would be helpful to compare and contrast these findings with data from other parts of the world. • The limitation section can be strengthened by adding information on data quality issues in dhis2 collection and reporting, and their generalizability. Overall, the paper provides a good insight on the utility of essential health services during Covid-19 in Kenya.
--

REVIEWER	Asar, Özgür Acibadem Universitesi
REVIEW RETURNED	12-Oct-2021

GENERAL COMMENTS	Quantifying the indirect impact of COVID-19 pandemic on utilisation of outpatient and immunisation services in Kenya: An interrupted time series analysis by Wamua et al. As requested I mainly review the statistical methods aspect of the paper. Below are the points that I would like to raise for the authors to address. More details are needed how regarding the multiple imputation, e.g. what are the mixed models and joint models used? The model equation in the section "Segmented regression" is actually for a linear regression, not for a generalised linear model. Can you write down the equation for the ARMA models and the expression for the Fourier terms. Presenting the results for different models, rather than only presenting the results for Negative-Binomial model, perhaps in a supplementary material would be nice. Why did you fit marginal models with generalised estimating equations?
---

VERSION 1 – AUTHOR RESPONSE

Reviewer: 1

Dr. Rossella Di Bidino, Fondazione Policlinico Universitario Agostino Gemelli IRCCS

Comments to the Author:

The paper is clear on methods, in the description of the local reality, and confounding factors (ie strike, missing data). Sensitivity analysis confirmed the robustness of your work,

I suggest minor revisions:

- Some information is duplicated in Methods on pre-e post COVID measures. Some of them are already reported in Introduction. Please, decide where to report it.

Thanks for identifying this, we have now excluded repeated sequence of events from introduction section and now only provided the information in methods section.

- Figure 3 is not easy to read. Have you considered the option to put it in Appendixes?

Thanks for noting this. We thought Figure 3 is important to present in the main text. This is because it shows exploratory trends of changes in services utilization of services as a first step towards a formal statistical analysis. It enabled us observe decline in services (in a general way) especially in December 2021, when there was a health workers' strike.

- In discussion, just introduce the point on potential impact of COVID 19 also on access to health services or mention it as point to investigate in future.

Thanks. That is a really important aspect to evaluate we agree. We alluded to this in the third paragraph of the discussion, where we identified reduced incomes affecting ability to meet transport costs and other healthcare related costs, shortage of healthcare workers in health facilities, difficulties in accessing health facilities due to lockdowns and curfew and closing of some health facilities as healthcare access factors contributing to the decline in utilization of services. These factors were identified in a survey https://preventepidemics.org/wp-content/uploads/2020/11/PERC-Brief-Essential-Services_Report_1120.pdf . We also hypothesised lockdowns restricted mobility and use of mobility data would help incorporate this and hence we further explored the COVID-19 stringency index for Kenya. (<https://ourworldindata.org/grapher/covid-stringency-index?tab=chart®ion=Africa&country=~KEN>), which is a composite measure of indicators like school closure, travel bans and is an indirect measure of mobility. The index indicated that the lockdown measures were most stringent around April 2020 for Kenya, the interruption point in our study. More information about composition and calculation of the index can be found here (<https://www.nature.com/articles/s41562-021-01079-8>) This has been added in line 192-193 with references.

Reviewer: 2

Dr. Arvind Kumar, All India Institute of Medical Sciences

Comments to the Author:

See file attached.

The manuscript may be accepted after minor revision. But for methodology section it requires statistical experts' review.

The authors have attempted to quantify the hindrance in the accessibility of the overall health facilities in Kenya health management systems due to the emergence of COVID-19. The study covers an important topic in the concerned area. However, the study requires revision before considering it for publication.

1) The abstract may be improved in terms of coverage of methodology part. The authors have used advanced statistical tools but they did not mention it in the abstract.

Thanks. We have now provided a summary of the statistical methods used to assess the changes in utilization of services. This has been added in the setting section of the abstract.

2) Similarly, the methodology part in the main text should also have comprehensive section over statistical tools used in the study.

Thanks. We have provided a comprehensive section on the interrupted time series model used for this study under "Interrupted time series analysis" section. We have now added a detailed explanation on the multiple imputation framework used to address missing data in the supplementary materials, Additional File 2.

3) In the methodology section, the independent variables and the dependent variables should be clearly defined for a thorough understanding of the readers.

Thanks for the observation. We have provided this information under the "Segmented regression" section. We have provided the interrupted time series model, and identified attendance in a given month as the dependent variable, while the independent variables are time data was reported, and a COVID-19 binary indicator (0 – before covid19 and 1 – after covid19) and a trend variable (time after covid19). Please see more on the section on the main text.

We have now added a section on multiple imputation model specification on Additional File 2. Indicating the variables used in the multilevel imputation model.

4) Use of multiple imputation for handling missing data was not clearly mentioned in the manuscript. If there were no cases reported that does not mean the misreporting of the data.

Thanks. Although the data is reported monthly, the chance that a health facility does not report a case (zero reports) in the month is possible. We therefore only included health facilities with more than 30% of months of data reported to ensure we analyse a cohort of health facilities that report consistently and ensure the imputations were more reliable and robust. The imputation was also carried out at the health facility level to ensure variability in monthly attendance volumes among health facilities is adjusted for. We also recognize that the data could be missing because no cases were reported because the health facility has been closed or has a duplicated record (a duplicated record where no data is reported). We therefore excluded all health facilities indicated to have been closed or duplicated (highlighted in Data section page 7).

We also included the time variable in the imputation method to incorporate changes in utilization of services in a given health facility for instance during the national strike.

Lastly, we have now suggested a sensitivity analysis study could be carried in future to assess any departure from the missing at random assumption when using multiple imputation in DHIS 2 data. Page 15, under Limitations and Recommendations section.

5) Also, if multiple imputation is required, the authors should clearly mention its use in details. Thanks. We used multiple imputation to adjust for missing data. Missing data is a confounder when assessing changes overtime. Also, we sought to adjust for missing data in line with a recent guide on using routine data to monitor the effects of COVID-19 by the WHO <https://www.who.int/bulletin/volumes/95/10/17-194399/en/> (Ref no. 70). We have provided more information under “Strengths and implications of the study”. Additionally, we used multiple imputation as it is the gold standard method for dealing with missing data (References provided on page 8). A recent study also identified multiple imputation to provide better performance for routine health information systems data in comparison to other studies <https://www.researchsquare.com/article/rs-422960/v1> . We have added this information now under the section on “Handling missing data”.

6) The results of the study are not self-explanatory. For example, headings of the figures and tables are not clearly mentioned by the authors.

Thanks for the observation. We have now provided more detailed explanations of the results in tables and figures headings. More detailed explanation on the tables and figures has also been updated in the legend

6) Overall, the structure of the manuscript should be revised and cab be spell checked and corrected for grammatical errors to make it more comprehensive.

Thanks. We have now amended and updated relevant sections. We have also added supplementary materials to further explain the study approaches in detail. We have also checked and corrected for grammatical errors.

Reviewer: 3

Dr. Subhash Chandir, Harvard Medical School Center for Global Health Delivery–Dubai

Comments to the Author:

The paper adds to the growing evidence on the impact of Covid-19 on essential health services. Their finds on the heterogeneous impact of Covid-19 on health services utility are exciting. Following are few suggestions for revision:

- Please consider adding references in "Pre-COVID-19 measures" and "Post-COVID-19 control measures" sections to support the narrative.

Thanks for the suggestion. We have now added references for the two sections.

- The study is based on dhis2 data. It would be helpful to provide information on the use of dhis2 system (reporting structure, frequency, data quality and reliability, monitoring, etc) to understand the comprehensiveness of data and its generalizability. There are many imputations, and although they are nicely carried out, please consider providing more context on the reason behind missing values.

Thanks for the observation. We have now added a section under Data section on "District Health Information Software version 2", where we have provided an explanation on the system in detail, page 7.

We have also added a section on missing data occurrence in DHIS 2 and some of the reasons behind missing data, page 8. But in general, missingness varies by health facility and consistency overtime. There are good reporting facilities regular reporting and some not so regular

- A significant finding is no impact on ANC1 visit while significant reduction is reported for ANC4 visit. However, ANC2 and 3 visits were excluded from the analysis. Please consider providing context, like was it due to lack of data or some other factor?

Thanks for the observation. ANC 1 and ANC 4 are recommended by WHO as tracker indicators for antenatal care coverage (<https://www.who.int/data/gho/indicator-metadata-registry/imr-details/80> and)and hence are the only antenatal care visits usually reported on DHIS 2. This has been clarified in main text on page 8, first paragraph

- The authors note there was a reduction in health visits for various reasons (pages 19 & 20). However, no explanation is given as to why the same deterrents did not affect the immunization visits. Moreover, it would be helpful to compare and contrast these findings with data from other parts of the world.

Thanks for the observation. We recognized unavailability of alternative sources for vaccination outside of the health system as a key factor in sustained immunization levels in Kenya during the pandemic. For instance, for outpatient conditions one can go to a local drug store/ self-medicate and do the same for their children, however this is not possible with vaccines as they are only available at health facilities/immunization centres.

Also in a recent survey across 18 African countries (including Kenya) <https://africacdc.org/download/using-data-to-find-a-balance-disruption-to-essential-health-services-in-africa-during-covid-19/> , which evaluated disruption to essential health services in Africa during COVID-19, found that vaccination was the least disrupted service across all countries, with Kenya not appearing in the list of countries where immunization was most affected. This information has now been added in the second paragraph of the discussion section.

In terms of comparing and contrasting the immunization results, we have also provided a detailed comparison of our results and other studies on immunization during COVID-19 in the second paragraph of discussion as well.

- The limitation section can be strengthened by adding information on data quality issues in dhis2 collection and reporting, and their generalizability.

Thanks for the observation. Although we adjusted for missing data and strikes in our analysis, which are potential confounders in our study in regard to data quality, we also dealt with other data quality issues such as removing duplicated health facilities, those indicated as closed, and extreme outliers were treated as missing. This information has been provided under "Extracted data" subsection. We also identified as an implication of this study some recommended strategies in the literature to improve data quality in DHIS2 such as investment in better infrastructure, supervisory support, formal data quality assurance and human resources could improve reporting in Kenyan health facilities in the strengths and implications section.

Overall, the paper provides a good insight on the utility of essential health services during Covid-19 in Kenya.

Thanks for the observation. We appreciate your time and input to better the study.

Reviewer: 4

Dr. Özgür Asar, Acibadem Universitesi

Comments to the Author:

Quantifying the indirect impact of COVID-19 pandemic on utilisation of outpatient and immunisation services in Kenya: An interrupted time series analysis by Wamua et al.

As requested I mainly review the statistical methods aspect of the paper. Below are the points that I would like to raise for the authors to address.

More details are needed how regarding the multiple imputation, e.g. what are the mixed models and joint models used?

Thanks. We agree and have provided detailed information on the multiple imputation framework used in Additional File 2 and referenced in the "Handling missing data section"

The model equation in the section "Segmented regression" is actually for a linear regression, not for a generalised linear model. Can you write down the equation for the ARMA models and the expression for the Fourier terms. Presenting the results for different models, rather than only presenting the results for Negative-Binomial model, perhaps in a supplementary material would be nice.

Thanks, the segmented regression equation has been updated to reflect the generalized linear regression model we used.

The ARMA model equation has now been provided in the second paragraph of the “segmented regression” section. We used the Gaussian copula marginal regression (gcmr) R package to fit the ARMA model, we have referenced this package as well in the main text. We carried out various combinations of (p, q) and selected the combination that resulted to lowest AIC. We also note that autocorrelation only occurred for one indicator, ANC 1, and the ARMA model was only fitted for this indicator. The ARMA parameters for the model fitted for this indicator have been provided in Table 1 footnote as well.

We used pairs of sine and cosine functions as Fourier terms by specifying the number of sin and cosine pairs to include as 2 and

and the length of the period as 12 months as recommended by Bernal et al 2016

<https://pubmed.ncbi.nlm.nih.gov/27283160/> . This has been

We have now clarified we tested two models, a negative binomial model accounting for seasonality and another without accounting for seasonality. We selected the negative binomial models because the health facility level variations in attendance were high for each indicator as shown by the intraclass correlation coefficients (ICC) provided in Additional File 4 Table 1. The results for the model where seasonality was not adjusted have now been provided in Additional File 4 Table 3. We have also added the AIC values showing the model accounting for seasonality was the best fitting model. (This information has now been updated and additional file referenced in the second paragraph of “segmented regression section”).

Why did you fit marginal models with generalised estimating equations?

We used a GEE as a sensitivity model to test the effect of varying model assumptions, for instance any hospital-specific differences, on the estimates similar to Siedner et al 2020

(<https://bmjopen.bmj.com/content/10/10/e043763>).

VERSION 2 – REVIEW

REVIEWER	Kumar, Arvind All India Institute of Medical Sciences, Department of Medicine
REVIEW RETURNED	27-Dec-2021
GENERAL COMMENTS	The authors have done a commendable job in improving the manuscript. This version of the manuscript seems revised and well structured.
REVIEWER	Chandir, Subhash Harvard Medical School Center for Global Health Delivery–Dubai
REVIEW RETURNED	22-Dec-2021
GENERAL COMMENTS	Thank you for sharing the revision and responses.